# Enhanced Tribodegradation of a Tetracycline Antibiotic by Rare-Earth-Modified Zinc Oxide

**DOI:** 10.3390/molecules29163913

**Published:** 2024-08-19

**Authors:** Dobrina Ivanova, Hristo Kolev, Bozhidar I. Stefanov, Nina Kaneva

**Affiliations:** 1Laboratory of Nanoparticle Science and Technology, Department of General and Inorganic Chemistry, Faculty of Chemistry and Pharmacy, University of Sofia, 1164 Sofia, Bulgaria; dobrina.k.ivanova@gmail.com; 2Institute of Catalysis, Bulgarian Academy of Sciences, Acad. G. Bonchev St., bl. 11, 1113 Sofia, Bulgaria; hgkolev@ic.bas.bg; 3Department of Chemistry, Faculty of Electronic Engineering and Technologies, Technical University of Sofia, 8 Kliment Ohridski Blvd, 1756 Sofia, Bulgaria; b.stefanov@tu-sofia.bg

**Keywords:** tribocatalysis, zinc oxide powder, rare earths, doxycycline, water remediation

## Abstract

Tribocatalysis is an emerging advanced oxidation process that utilizes the triboelectric effect, based on friction between dissimilar materials to produce charges that can initiate various catalytic reactions. In this study, pure and rare-earth-modified ZnO powders (La_2_O_3_, Eu_2_O_3_, 2 mol %) were demonstrated as efficient tribocatalysts for the removal of the tetracycline antibiotic doxycycline (DC). While the pure ZnO samples achieved 49% DC removal within 24 h at a stirring rate of 100 rpm, the addition of Eu_2_O_3_ increased the removal efficiency to 67%, and La_2_O_3_-modified ZnO powder exhibited the highest removal efficiency, reaching 80% at the same stirring rate. Additionally, increasing the stirring rate to 300 and 500 rpm led to 100% DC removal in the ZnO/La case within 18 h, with the pronounced effect of the stirring rate confirming the tribocatalytic effect. All tribocatalysts exhibited excellent recycling properties, with less than a 3% loss of activity over three cycles. Furthermore, a scavenger assay confirmed the importance of superoxide radical generation for the overall reaction rate. The results of this investigation indicate that the rare-earth-modified ZnO tribocatalysts can effectively utilize mechanical energy to decompose pollutants in contaminated water.

## 1. Introduction

The major environmental hazard worldwide and a significant contributor to the decline in human health is wastewater, due to its complex composition and slow degradation rate [1,2,3]. Degrading organic substances in sewage using widely available environmental energy sources, such as solar, thermal, and mechanical energy, is a promising strategy for environmental remediation [4,5,6,7]. Among these, photocatalysts can absorb solar energy, one of the cleanest and most renewable energy sources, to initiate various photocatalytic reactions, including the reduction of carbon dioxide and water splitting for hydrogen production [8,9]. Photocatalysis can also be employed to remediate polluted wastewater under UV or visible light irradiation for the degradation of organic pollutants. However, photocatalysis technology has limited practical applications due to the high recombination rate of photo-excited carriers and challenges in transmitting light through optically opaque environments.

A new avenue to energy-catalyzed organic pollutant removal is pyroelectric catalysis, where heat energy is used for charge generation. However, most pyroelectric catalysts require a high rate of temperature change to efficiently convert thermal energy into chemical energy, and meeting the excitation conditions of thermal catalysis in aqueous environments is challenging [10]. Therefore, developing a novel catalytic approach to address current environmental pollution issues is imperative.

This leads to the possibility of utilizing mechanical energy for catalytic conversion. In this context, the tribocatalytic effect has attracted significant interest due to its repeatability and environmentally friendly nature [11,12,13]. Tribocatalysis is based on the triboelectric effect, where triboelectric charges are generated when two materials come into contact through friction [14,15,16]. A promising method to reduce environmental pollution is to use the positive and negative triboelectric charges produced during friction to react with oxygen and hydroxyl ions in water, respectively, for reactive species generation which may in turn be employed to decompose hazardous dyes in effluents from the textile industry [17,18]. For the first time, Li et al. reported that tribocatalytic Ba_0.75_Sr_0.25_TiO_3_ nanoparticles could effectively break down organic dyes in a liquid environment [19]. These nanoparticles absorb frictional energy between glass and PTFE surfaces, breaking down dye molecules through electron–hole pair activation and subsequent redox chemical reactions. Zhao et al. [20] employed ZnO nanorods and PTFE magnetic bar stirring to degrade Rhodamine B (RB) dye; at a rotation speed of 1000 rpm, the RB dye decomposition efficiency reached 99.8% within 60 h. Thus, utilizing the frictional effect of nanomaterials to degrade dyes has emerged as a novel concept; however, the degradation of drugs by ZnO-based tribocatalysts has not been widely studied. Only two publications, by Sun et al. [21] and Li et al. [22], have demonstrated the tribodegradation of tetracycline using alternative materials—FeOOH nanorods and pyrite-based tribocatalysts, respectively.

In this study, ZnO was chosen due to its remarkable semiconductor properties, high chemical stability, environmental friendliness, and piezoelectric properties, which are key for the fabrication of efficient tribocatalysts. Pure and La- or Eu-modified ZnO powders were prepared to examine their tribocatalytic efficiency in decomposing doxycycline (DC), motivated by the expectation that rare-earth elements, known to enhance piezoelectric properties, would improve the triboelectric efficiency of the ZnO catalyst. We demonstrate that doping with rare-earth ions and increasing the magnetic stirring speed enhances degradation activity, confirming that the degradation of DC is driven by mechanical energy. This finding aligns with our previous work, where we showed that rare-earth doping improves the photocatalytic efficiency of ZnO by suppressing the recombination of photogenerated electrons and holes [23].

## 2. Results and Discussion

### 2.1. Structural and Morphological Characterization of the Pure and Rare-Earth-Modified ZnO Tribocatalysts

The microstructure and morphology of the ZnO samples modified with Eu and La rare-earth (RE) ions were examined in detail using scanning electron microscopy (SEM). The SEM images (Figure 1) reveal that the ZnO nanocomposites consist of particles with different shapes and sizes. An analysis of the SEM images shows an average particle diameter of 0.7 ± 0.1 µm for pristine ZnO, which remains consistent in the ZnO/Eu sample (0.7 ± 0.2 µm) and increases slightly to 0.8 ± 0.2 µm in the ZnO/La case. Despite the incorporation of Eu_2_O_3_ and La_2_O_3_ in the ZnO/Eu and ZnO/La case, respectively, the low treatment temperature (100 °C) preserved the morphology of the samples, with no significant effect observed based on the type of rare-earth element.

The Brunauer–Emmett–Teller (BET) surface area analysis revealed that all the RE-modified samples exhibited a higher surface area than pure ZnO (10.3 ± 1.6 m^2^/g). Notably, the ZnO/La powder displayed the largest surface area (32.3 ± 1.8 m^2^/g), suggesting the potential for higher tribocatalytic activity compared to ZnO/Eu (30.3 ± 1.7 m^2^/g).

Energy-dispersive X-ray spectroscopy (EDS) confirmed the presence of Zn, O, and rare-earth elements in the modified ZnO powders (Figure 2). Peaks corresponding to zinc, oxygen, and the respective rare-earth elements were observed. The RE elements’ weight percentage was approximately 3 wt. % in all cases (Table 1). The absence of impurity peaks in the EDS spectrum indicates the high purity of the starting ZnO material. Europium and lanthanum were homogeneously distributed across the ZnO surface, as evidenced by the mapping data in Figure 2c.

Transmission electron microscopy (TEM) further revealed the morphology and positioning of the RE-oxide phase on the ZnO surface, with the resulting micrographs presented in Figure 3.

The TEM images confirmed that the tribocatalysts consist of polycrystalline agglomerates several hundred nanometers in diameter, consistent with SEM observations. The ZnO phase is decorated by the respective Eu_2_O_3_ and La_2_O_3_ co-catalyst RE-oxide phases, which appear as surface-bound particles approximately 10 nm in diameter in higher magnification images (Figure 3c,d).

The chemical state of the RE-modified ZnO tribocatalysts was investigated via X-ray photoelectron spectroscopy (XPS). Figure 4a shows the Zn 2p region of the XPS spectrum, where a peak with a binding energy (BE) of 1021.7 eV and another band at 1045 eV correspond to Zn 2p_1/2_ and Zn 2p_3/2_ in ZnO.

The O 1s region (Figure 4b) shows a doublet at 530.3 eV and 531.6 eV, corresponding to extraneous and lattice oxygen in wurtzite ZnO [24]. No significant changes were observed in the Zn 2p and O 1s XPS spectra for ZnO/Eu (Figure 4c,d) and ZnO/La (Figure 4f,g). In the ZnO/Eu case (Figure 4e), a peak around 1135 eV BE corresponds to the Eu 3d_5/2_ level, indicating oxygen-coordinated Eu^3+^ [25]. For the ZnO/La sample, the La 3d region (Figure 4h) shows a doublet at 835.2 eV with a satellite at ~838 eV BE, consistent with the presence of La^3+^ [26].

XPS analysis revealed an RE concentration of 5.52 at. % for ZnO/La, suggesting uniform coverage of the ZnO particles with the dopant. However, for the ZnO/Eu case, only 0.24 at. % was detected via XPS, which is inconsistent with the EDX results. This discrepancy suggests that ZnO particles accumulate around the Eu_2_O_3_ oxide, masking it in the surface-sensitive XPS analysis.

Powder X-ray diffraction (XRD) patterns of the pure and RE-modified ZnO powders after annealing at 100 °C are shown in Figure 5. The crystalline phase of pure ZnO is hexagonal wurtzite, evidenced by intense diffraction peaks at 2θ = 31.94°, 34.67°, 36.51°, 48.23°, 56.84°, 63.22°, 67.53°, and 68.18°. These peaks correspond to the lattice planes (100), (002), (101), (102), (110), (103), (112), and (201), respectively [27]. The peak positions align with JCPDS Card No. 36-1451. No impurities or phase modifications were observed in the crystalline structure, and the XRD patterns of both pure and RE-modified ZnO samples show strong, sharp peaks, indicating a high degree of crystallinity [28]. The RE^3+^ phase is uniformly distributed as tiny oxide clusters among ZnO nanoparticles, and the low concentration of lanthanide ions (2 mol %) in the modified ZnO composite may explain the absence of a distinct phase. Notably, the crystallite size of RE-modified ZnO is larger than that of pure ZnO, as determined by the Scherrer equation using the main peak (101). The crystallite size of pure ZnO is 37 nm, while the modified ZnO catalysts have crystallite sizes of approximately 42 nm. The increase in the crystallite size of ZnO/RE could be attributed to bond formation between the oxides on the surface of the composite samples, which affects the crystallite size [29,30].

The XRD data show no discernible change in crystal size with the addition of lanthanide ions (2 mol %). Except for a minor increase in the average crystallite size after modification, the crystalline lattice parameters remain largely unchanged (Table 2). The calculated lattice parameters closely resemble those of ZnO, indicating that powders modified with rare-earth elements maintain their hexagonal wurtzite structure. A positive value for tensile strain is observed when the microstrain of the samples is calculated using the c-axis lattice parameter. The tensile strain is slightly reduced in modified powders compared to ZnO.

Raman spectroscopy further confirmed the phase composition of the ZnO materials, with the spectra depicted in Figure 6. In all cases, the most intensive Raman bands were observed at 331, 439, and 1154 cm^−1^, corresponding to ZnO’s E_2_(high −)–E_2_(low) mode, the E_2_(high) mode, and the 2A_1_(low) + 2E_2_(low) broad band, respectively [31]. These data are consistent with wurtzite ZnO and do not suggest any major modifications of the main tribocatalyst component due to functionalization.

### 2.2. Tribocatalysis for Decomposition of Doxycycline—Effect of Rare-Earth Elements

The tribocatalytic efficiency of pure and rare-earth-modified ZnO powders was evaluated for the degradation of a tetracycline antibiotic, doxycycline (DC), under dark conditions. Magnetic stirring facilitated the tribocatalytic process, and the drug concentration was consistently maintained at 15 mg/L in all experiments. UV/Vis spectroscopy was employed to monitor the degradation of DC by tracking the absorption maxima at 275 nm. To determine how the different rare-earth elements affect ZnO’s activity during the tribocatalytic process, spectral changes in the degradation of DC were examined. Figure 7 presents the UV/Vis spectra for DC degradation using pure ZnO, ZnO/Eu, and ZnO/La, respectively.

Figure 8a displays the tribodegradation results (at 300 rpm), highlighting that ZnO/La powder exhibits the fastest degradation rate, achieving 92.5% degradation after 24 h of friction. In contrast, a control experiment without a tribocatalyst showed negligible degradation (~4%), underscoring the importance of the catalyst in the friction process. The catalytic efficiencies follow the order ZnO/La > ZnO/Eu > ZnO. The higher efficiency of ZnO/La is likely due to its increased specific surface area, which promotes better separation of tribogenerated electron–hole pairs and enhances carrier participation in redox reactions, along with providing an increased number of active sites for DC adsorption [32]. The reaction rate constants, as shown in Figure 8b, were determined by ln(C_t_/C_0_) = −kt pseudo-first order kinetics, typically used to describe photo- and tribocatalytic removal [20] and to further confirm this trend, with ZnO/La exhibiting the highest rate constant (k = 0.1015 h^−1^).

### 2.3. Tribocatalysis for Decomposition of Doxycycline—Plausible Mechanism

The mechanism of tribocatalysis is still being explored, but two primary pathways have been proposed [33]: (i) electron transfer from the tribocatalyst to the PTFE bar; and (ii) excitation of electron–hole pairs due to ZnO deformation, similar to photocatalysis [34]. Figure 9 illustrates a schematic representation of tribocatalysis by ZnO and ZnO/RE composites.

During rotational friction stirring, the interaction between the PTFE bar and tribocatalyst powders generates positive and negative charges through electron extraction onto the PTFE surface. PTFE absorbs electrons from the ZnO and ZnO/RE surfaces and, additionally, electron–holes are generated in the semiconductor chemical reactions caused with the DC molecule. Electrons represent excited e^−^, and holes represent the formed h^+^ that results from ZnO absorbing mechanical energy during friction; similarly, heterogeneous photocatalysis in which organic pollutants are broken down by photoexcitation of ZnO electron–hole pairs is comparable to frictional contact-induced catalysis [35,36]. During the decomposition of the drug, the oxygen molecules react with the tribogenerated electrons on the PTFE surface, and superoxide radicals are formed, while the holes remaining on the ZnO surface may interact with OH^−^ and be transformed into OH^•^, with both tribogenerated radicals effectively attacking the DC molecule.

Two fundamental questions must be addressed to comprehend the energy transfer in RE-modified ZnO: (i) How does the rare-earth ion energy level relate to the host tri-bocatalyst’s valence and conduction bands? (ii) How do the locations of rare-earth ion energy levels impact the processes of charge migration and trapping?

Duffy’s oxide model [37] is thus used to calculate the energy band gaps of ZnO and RE ions as a function of their optical electronegativity: E_g_ = 3.71 × ∆χ, where ∆χ is the optical electronegativity of the binary oxide, which is 3.15 for ZnO, 2.54 for Eu_2_O_3_ and 2.5 for La_2_O_3_. The calculated band gap values are discovered to increase in the following order: E_g[ZnO]_ = 3.3 eV < E_g[ZnO/Eu]_ = 4.3 eV < E_g[ZnO/La]_ = 5.5 eV. To confirm this expectation, UV/Vis diffuse reflectance spectra were obtained for the pure ZnO and ZnO/RE composites, and then converted using the Kubelka–Munk approach: F(R) = (1-R)^-2^/2R, where F(R) is the Kubelka–Munk (K-M) function, which can be approximated functionally to an absorption coefficient, i.e., F(R)∝α, and used directly to obtain the optical bandgap (E_g_) of the powders via Tauc analysis. As ZnO is a direct bandgap semiconductor, E_g_ can be obtained as the cross-section of the functional dependence (F(R)hυ)^2^ vs. hυ, where hυ is the energy corresponding to the wavelength at which the reflectance value of the obtained F(R) was measured. Figure 10 depicts the resulting Tauc plots for ZnO and the three ZnO/RE composites.

As shown in Figure 10, the experimentally observed optical bandgaps closely follow the expected arrangement, predicted by Duffy’s model: E_g[ZnO]_ < E_g[ZnO/Eu]_ < E_g[ZnO/La]_; however, only a modest difference of 0.05 eV is observed across the ZnO and ZnO/La case. It should be noted, however, that in the model case, a binary oxide is assumed, while as seen by the TEM evidence in Figure 3, a heterostructure between ZnO and RE oxide is formed; hence, the optical absorption will be governed mainly by the ZnO semiconductor. As the most active tribocatalyst, namely ZnO/La, also exhibits the highest bandgap, it could be suggested that the main contribution to its enhanced activity is expected to be the improved triboelectric charge separation between the ZnO and the RE oxide phase and the formation of a heterojunction between the two dissimilar semiconductors, which has been demonstrated as an effective strategy for improved tribocatalytic activity in the literature [38]. Additionally, the RE ion’s 4f-shell can take electrons from or transfer them to the energy bands of a compound through the RE^3+^/RE^2+^ or RE^4+^/RE^3+^ valence change, in which, as illustrated in Figure 9, the La and Eu ions in ZnO composite catalysts could contribute to electron trapping and transfer.

Apart from the enhanced activity of the La and Eu modification potentially attributed to electron–hole separation, the generation of O_2_^•−^ and OH^•^ radicals is among the main mechanisms in tribocatalysis [18,38]. The highest efficiency is seen in the La-modified ZnO sample, which can be guessed as possibly being due to the higher number of oxygen vacancies in this instance (caused by the differing charge and electronegativity of lanthanum and zinc ions) and the stronger hydroxyl ion adsorption onto the ZnO surface [39]. The reaction between the hole and OH^−^ promotes the formation of OH^•^. Degradation of the organic pollutant at the surface of La-modified ZnO can thus be linked to the tribogeneration of the extremely potent non-selective oxidants OH^•^ [40]. Since the added RE energy levels in the case of Eu-modified ZnO are near but below the energy of the ZnO conduction band, the reaction of the tribogenerated electron and O_2_ molecules favors the formation of O_2_^•−^ radicals.

Introducing the RE oxide phase into ZnO creates distinct energy levels and potentially suppresses tribogenerated charge recombination, further boosting catalytic efficiency. The RE phase helps to trap electrons, prevent electron–hole recombination, and produce more superoxide and hydroxyl radicals, all of which contribute to pollutant degradation.

To confirm the involvement of hydroxyl and superoxide radicals, a radical scavenger assay was performed using ascorbic acid (AA) and isopropyl alcohol (IPA) as scavengers for superoxide (O_2_^•−^) and hydroxyl (OH^•^), respectively [41,42].

Figure 11 shows that three tribocatalyst systems responded similarly to the addition of AA and IPA scavengers. The results show that superoxide radicals have a more significant impact on the DC tribodegradation rate in all three tribocatalyst systems, as evidenced by the pronounced inhibition with AA.

### 2.4. Tribocatalysis for Decomposition of Doxycycline—Effect of Magnetic Stirring and Catalyst Recycling

The impact of varying rotational speeds on the tribocatalytic breakdown of doxycycline (DC) using pure and RE-modified ZnO powders was studied at different speeds: 100, 300, and 500 rpm, and as illustrated in Figure 12 there is a concomitant increase in DC removal with stirring speed. The pure ZnO sample exhibited degradation efficiencies of 49.2%, 66.7%, and 80.4% at 100, 300, and 500 rpm respectively.

The ZnO/La sample demonstrated the highest catalytic performance, achieving 100% drug degradation in less than 20 h at a rotation speed of 500 rpm, outperforming the other modified samples (Figure 12a). In all cases, the tribocatalytic activity followed the order of ZnO/La > ZnO/Eu > ZnO and the increased rotation speed enhanced the rate of drug decomposition (Figure 12b,c).

Table 3 presents the rate constants and percentages of DC decomposition after the first tribocatalytic cycle. The data support the conclusion that rare-earth elements enhance the tribocatalytic process, allowing for effective drug degradation even in the absence of light.

Figure 13 shows the results of a study on the recyclability of ZnO, Eu/ZnO, and La/ZnO powders over three consecutive cycles. The catalytic properties of the powders declined slightly with each cycle, with the tribocatalytic degradation of DC decreasing by approximately 3% for each type of catalyst after three cycles in distilled water. Despite this decrease, the hydrothermal powders demonstrated good cycling stability for DC decomposition. Notably, the ZnO/La nanostructures maintained the highest degree of tribocatalytic activity across all cycles, confirming their potential for repeated use in DC degradation. These findings indicate that while the tribocatalytic efficiency of the powders decreases slightly with repeated use, the ZnO/La composite remains the most effective and stable catalyst over multiple cycles.

## 3. Materials and Methods

### 3.1. Reagents and Preparation of RE-Modified ZnO Powders

Zinc oxide commercial powder (>99.0%), La_2_O_3_ (>99.0%), Eu_2_O_3_ (>99.0%), and absolute C_2_H_5_OH were obtained from Fluka, Burlington, MA, USA).

Doxycycline (C_22_H_24_N_2_O_8_, λ_max_ = 275 nm, Teva, Sofia, Bulgaria) was selected for the tribocatalytic experiments as the modal pollutant because of its widespread use in real-world settings.

A straightforward and environmentally friendly hydrothermal process was used to create three series of ZnO/RE composite powders. In a glass vessel, the appropriate amounts of commercial ZnO powder and La_2_O_3_ (2 mol %) were combined, and ethanol was added as a mixing medium to create La-modified tribocatalysts. The materials were combined for ten minutes, sonicated for thirty more minutes, and then dried for one hour at 100 °C to produce the ZnO/La powders needed for tribocatalytic testing. The remaining catalyst (Eu) was prepared under the same ideal conditions, with a concentration of 2 mol% of RE ions.

### 3.2. Instrumental Methods

The surface morphology of pure ZnO and ZnO modified by RE was examined using SEM (JSM-5510, Krefeld, Germany) operating at 10 kV of acceleration voltage. For elemental analysis or chemical characterization of the samples, energy-dispersive X-ray spectroscopy (EDXdetector: Quantax 200, Bruker Resolution 126 eV, Berlin, Germany) was employed. Transmission electron microscopy was performed on a JEOL JEM-2100 (JEOL Ltd., Tokyo, Japan), operating at 200 kV. Based on Brunauer–Emmett–Teller (BET) N_2_ absorption (Quantachrome Instruments NOVA 1200e, Boyton Beac, FL, USA), the surface area of pure and RE/ZnO composite powders was estimated. The samples were degassed at 150 °C for four hours before N_2_ adsorption for the BET analysis. XRD (Siemens D500 with Cu Kα radiation, Karlsruhe, Germany) was used to analyze the crystallinity and phase composition of the catalysts. Scherrer’s equation was used to estimate the average crystallite sizes. X-ray photoelectron measurements were performed using the ESCAAB MkII electron spectrometer (VG Scientific, now Thermo Scientific, Manchester, UK) equipped with a twin anode MgKα/AlKα non-monochromated X-ray source that used excitation energies of 1253.6 and 1486.6 eV, respectively. The base pressure in the analysis chamber was 5 × 10^−10^ mbar. The only non-monochromated X-ray source used for the measurements was AlKα. There was roughly 1 eV in the instrumental solution. SpecsLabl2 CasaXPS software (2.3.25PR1) was used to analyze the data. Shirley-type background and X-ray satellite subtraction were used in the processing of the measured spectrum. Using a symmetric Gaussian–Lorentzian curve fitting, the peak positions and areas were assessed. Raman spectrometry was carried on a ThunderOptics Eddu TO-ERS-532 spectrometer, Montpellier, France), equipped with a 532 nm laser source and a 20× microscope objective lens. UV/Vis spectra were obtained on an Evolution 300 Thermo Scientific spectrophotometer (Madison, WI, USA), equipped with a DRA-EV-300 Diffuse Reflectance Accessory.

### 3.3. Tribocatalytic Experiments and Radical Assay

The tribocatalytic experiments were carried out with 50 mL DC solution prepared with distilled water in a 100 mL glass beaker, equipped with a magnetic stirrer. The tribocatalytic reaction was conducted at constant room temperature (23 ± 2 °C) in the dark. The initial concentration of DC was 15 ppm. In total, 50 mg catalyst (pure or RE-modified ZnO) was added to a glass reactor containing DC solution and the suspension was magnetically stirred using a PTFE-coated magnetic bar (ø 8 mm, L = 35 mm). To attain the adsorption equilibrium between the doxycycline solution and tribocatalysts, the resultant mixture was soaked for 30 min without any magnetic stirring. After that, the reactor was turned on, initially rotating at a constant speed of 300 rpm. At regular intervals, aliquot samples of 2 mL of the reaction solution were taken. The tribocatalyst was then centrifuged at 6000 rpm. UV–Vis spectra of aliquots from the reaction media were recorded in the range of 200–450 nm. The peak at maximal drug absorption was at 275 nm (absorbance decreased as a function of stirring time for each catalyst). At this wavelength, not only was the degradation of doxycycline observed but also its degradation products (phenolic compounds) [43,44]. This method was similar to all other decomposition performance tests, except for differences in the type of catalyst (pure and ZnO/RE powders) and magnetic stirring conditions (100 and 500 rpm).

The following formula was used to estimate the drug tribodegradation degree (D%):(1)$$Decomposition\%=\frac{C}{C_{0}}\times 100\%$$
where C_0_ is the initial concentration of doxycycline and C is the concentration (absorbance) of drug at time = t (min) [18,22].

Additionally, blank experiments without catalysts were carried out—there was no sign of any removal of the tetracycline antibiotic under PTFE stirring without a tribocatalyst.

The reactive species causing the degradation of the DC were investigated using a scavenger test. Isopropyl alcohol (IPA) and ascorbic acid (AA) were used as scavengers to absorb superoxide and hydroxyl radicals, respectively. To identify the specific reactive species that underwent tribocatalysis-induced degradation of the organic dye (50 mL), 6 mM of each scavenger was used separately.

## 4. Conclusions

In this study, the tetracycline antibiotic doxycycline (DC) was successfully degraded using magnetic stirring in the dark, facilitated by three types of ZnO powders modified with rare-earth elements (Eu^3+^ and La^3+^). Among these, the ZnO/La composite, which exhibited the highest specific surface area, demonstrated the most effective degradation. The degradation rate significantly increased with higher stirring speeds, emphasizing the role of mechanical energy in the process. The results reveal that mechanical energy absorbed during friction effectively excites the electrons and holes in ZnO and ZnO/RE composites, leading to efficient drug breakdown. This tribocatalytic effect represents a promising, eco-friendly pathway for harnessing mechanical energy from the environment to address pollution. By enabling drug degradation through tribocatalysis, this method opens new possibilities for controlling environmental contamination.

## Figures and Tables

**Figure 1 molecules-29-03913-f001:**
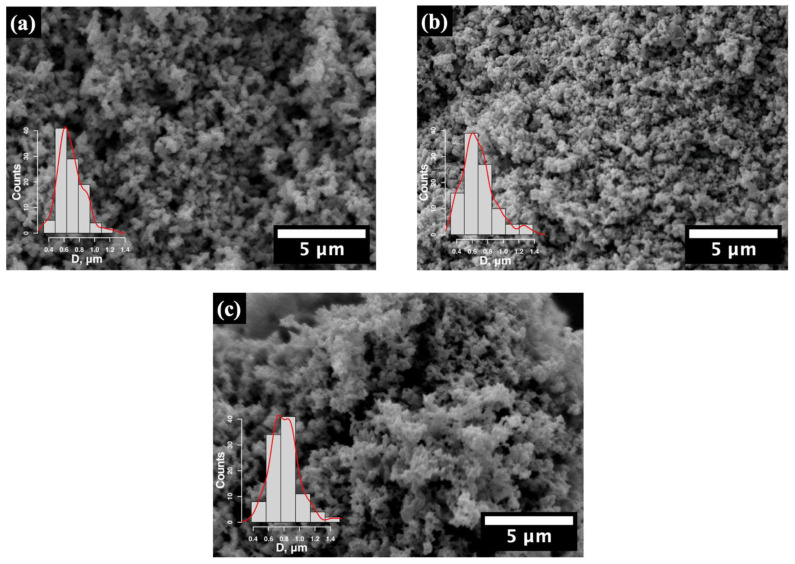
SEM micrographs of (**a**) pure ZnO; (**b**) Eu-modified ZnO; and (**c**) La-modified ZnO powder. Insets show the particle size distribution obtained from the respective micrographs.

**Figure 2 molecules-29-03913-f002:**
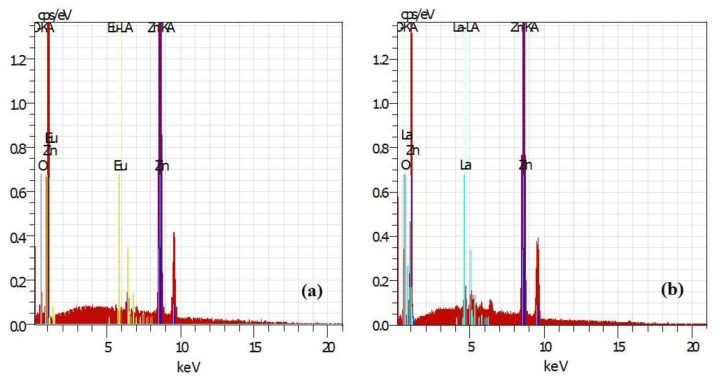
EDS spectra of ZnO powders modified with different rare-earth ions (2 mol %): (**a**) Eu; (**b**) La; (**c**) mapping surface data for rare-earth-modified samples.

**Figure 3 molecules-29-03913-f003:**
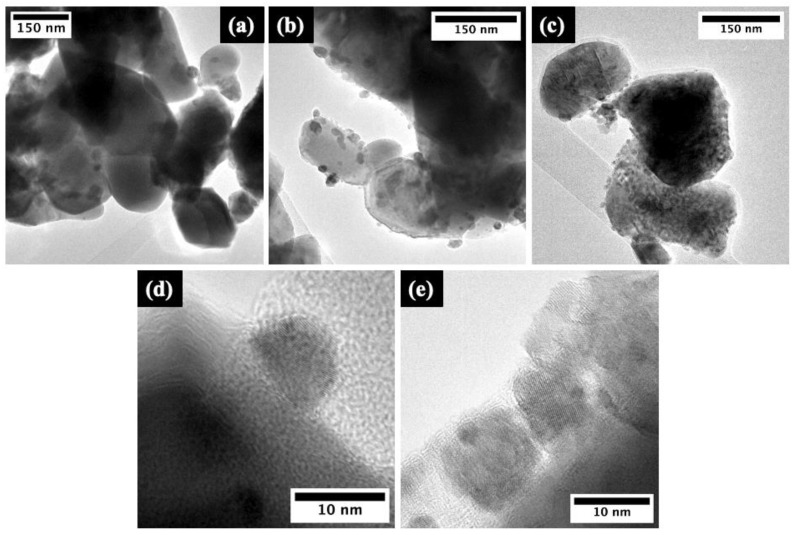
TEM micrographs of (**a**) pure ZnO, (**b**,**d**) ZnO/Eu, and (**c**,**e**) ZnO/La powders.

**Figure 4 molecules-29-03913-f004:**
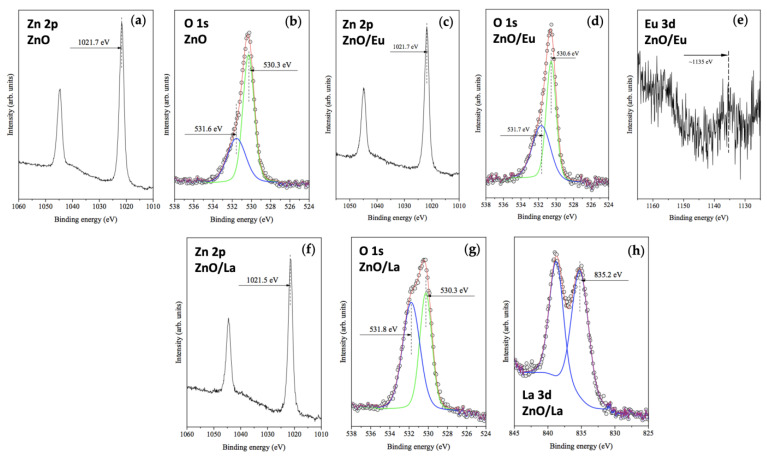
XPS spectra of ZnO and RE-modified ZnO powders showing (**a**) the Zn 2p region in pure ZnO; (**b**) the O 1s region in pure ZnO; (**c**) the Zn 2p region in ZnO/Eu; (**d**) the O 1s region in ZnO/Eu; (**e**) the Eu 3d region in ZnO/Eu^3+^; (**f**) the Zn 2p region in ZnO/La; (**g**) the O 1s region in ZnO/La; and (**h**) the La 3d region in ZnO/La.

**Figure 5 molecules-29-03913-f005:**
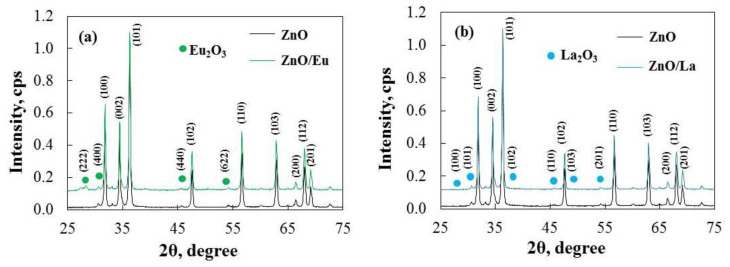
XRD patterns of (**a**) ZnO/Eu and (**b**) ZnO/La powders, with main reflections denoted, according to JCPDS 36-1451 (wurtzite ZnO), JCPDS 43-1008 (Eu_2_O_3_), and JCPDS 83-1355 (La_2_O_3_).

**Figure 6 molecules-29-03913-f006:**
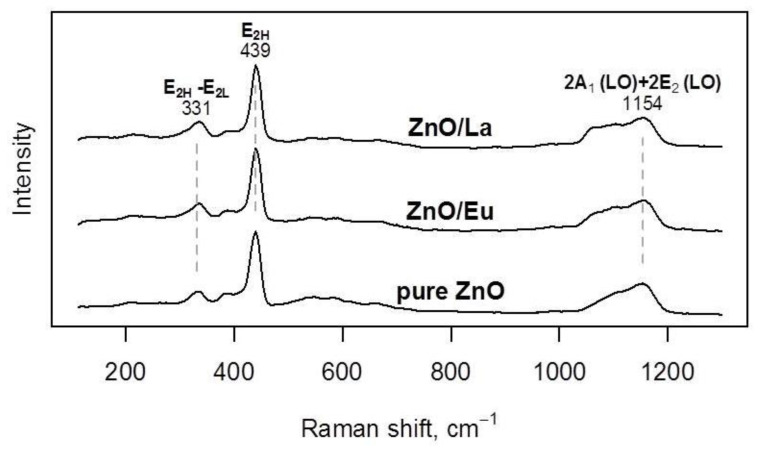
Raman spectra of pristine ZnO, ZnO/Eu, and ZnO/La powders.

**Figure 7 molecules-29-03913-f007:**
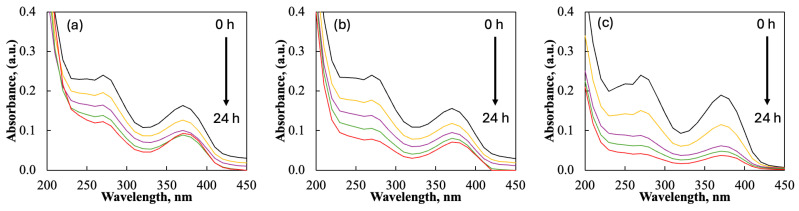
Spectral data for the tribodegradation of DC for (**a**) pristine ZnO; (**b**) Eu-modified ZnO; and (**c**) La-modified ZnO.

**Figure 8 molecules-29-03913-f008:**
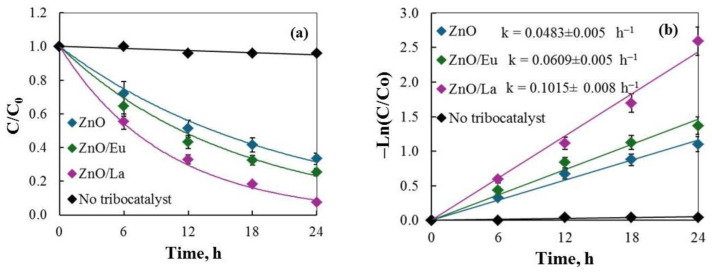
(**a**) Stirring degradation of DC solution using ZnO and ZnO modified with different rare-earth elements (2 mol%) under magnetic stirring conditions (300 rpm); (**b**) kinetic fitting.

**Figure 9 molecules-29-03913-f009:**
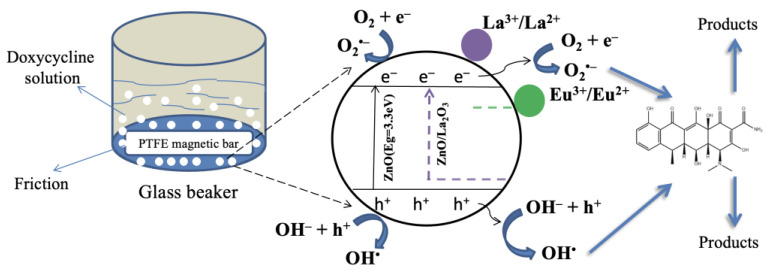
Plausible mechanism of the tribocatalysis degradation of DC by ZnO and ZnO/RE powders by analogy to [33,34].

**Figure 10 molecules-29-03913-f010:**
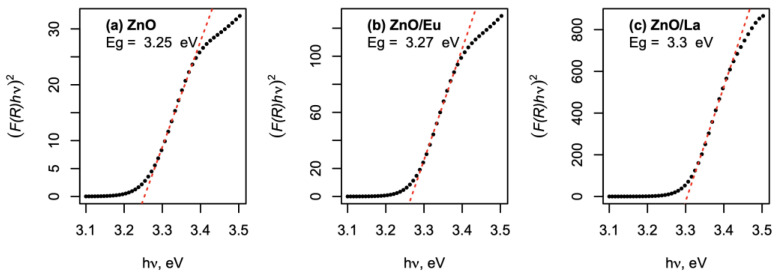
Tauc plots for (**a**) ZnO, (**b**) ZnO/Eu, and (**c**) ZnO/La tribocatalyst powders. The optical bandgap value is presented as an insert in each plot.

**Figure 11 molecules-29-03913-f011:**
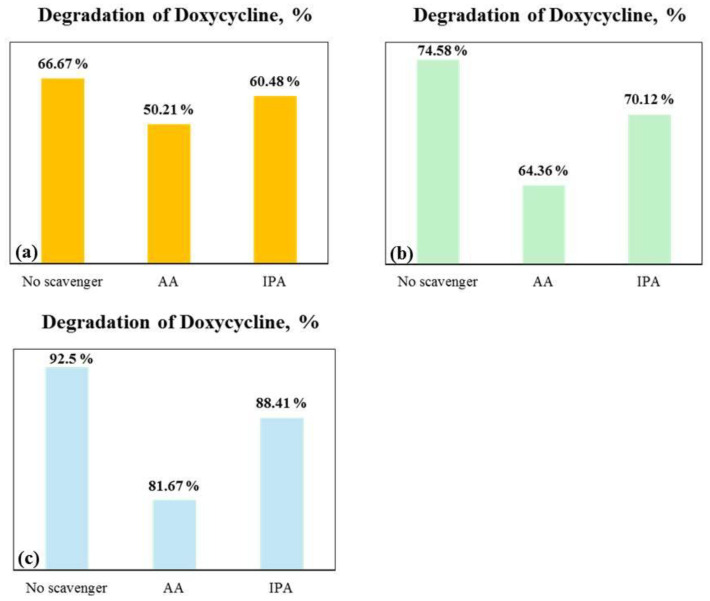
Effect of scavengers on DC degradation in tribocatalysis process using (**a**) pure, (**b**) europium, and (**c**) lanthanum-modified ZnO powder.

**Figure 12 molecules-29-03913-f012:**
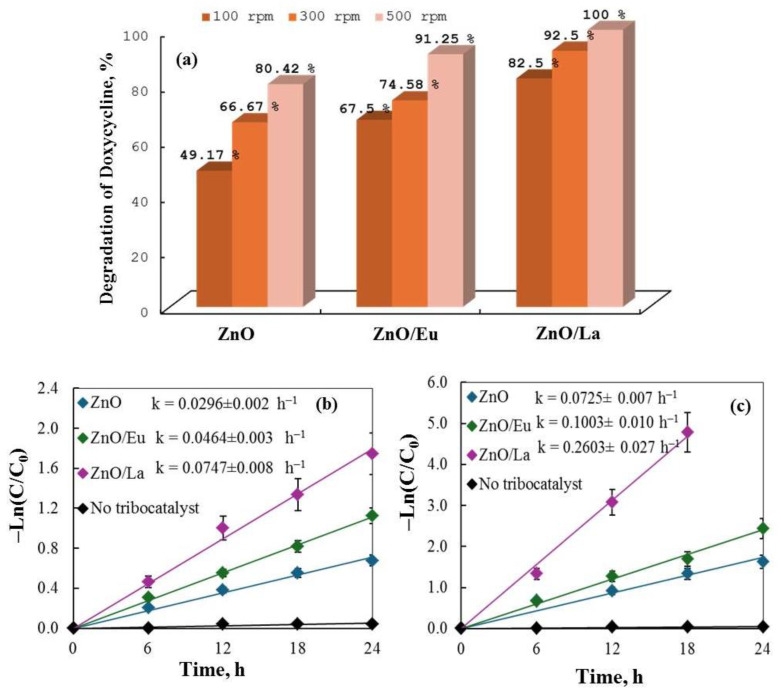
(**a**) Degradation of DC at different rotation speeds using pure and RE-modified ZnO composites; (**b**,**c**) ln(C/Co) vs. plot showing the rate of drug decomposition in tribocatalysis experiments by semiconductors stirred at 100 and 500 rpm.

**Figure 13 molecules-29-03913-f013:**
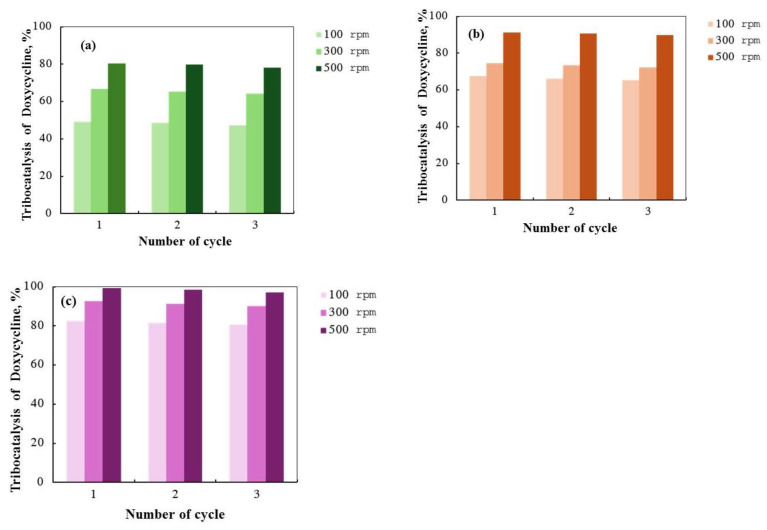
Tribocatalytic degradation rate of DC for three consecutive cycles using ZnO (**a**), Eu/ZnO (**b**), and La/ZnO (**c**) powders at different rotation speeds.

**Table 1 molecules-29-03913-t001:** EDS values of ZnO/RE powders.

Sample Powders	C Norm. [wt. %]		C Atom. [at. %]		C Error, [%]	
ZnO/Eu	O	41.66	O	64.90	O	4.9
	Zn	45.84	Zn	34.26	Zn	1.6
	Eu	3.25	Eu	0.84	Eu	0.1
ZnO/La	O	42.45	O	61.81	O	4.4
	Zn	54.28	Zn	36.45	Zn	1.5
	La	3.27	La	1.74	La	0.2

**Table 2 molecules-29-03913-t002:** Crystallite size and lattice parameters of pure and RE-modified ZnO powders.

Sample Powders	Crystallite Size, nm	Parameters of the Crystalline Lattice, Å	Microstrains, a.u.
ZnO	37	a, b: 3.2531 c: 5.2057	6 × 10^−4^
ZnO/Eu	41	a, b: 3.2516 c: 5.1535	4 × 10^−4^
ZnO/La	42	a, b: 3.2504 c: 5.1524	4 × 10^−4^

**Table 3 molecules-29-03913-t003:** The values of rate constants and percent of DC decomposition using the tribocatalytic process after the first cycle.

Sample Powders	100 rpm		300 rpm		500 rpm	
	k, h^−1^	D, %	k, h^−1^	D, %	k, h^−1^	D, %
ZnO	0.0296	49	0.0483	67	0.0725	80
ZnO/Eu	0.0464	68	0.0609	75	0.1003	91
ZnO/La	0.0747	83	0.1015	93	0.2603	100

## Data Availability

Data are contained within the article.

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
