# Peer review of "Enhanced Tribodegradation of a Tetracycline Antibiotic by Rare-Earth-Modified Zinc Oxide"

_molecules, 2024, doi:10.3390/molecules29163913_

Round 1
Reviewer 1 Report
Comments and Suggestions for Authors
Manuscript should be revised accordingly.

Requires sentence restructuring. Check for grammatical and typo errors.
Author Response
The manuscript: “Enhanced tribodegradation of a tetracycline antibiotic by rare-earth modified zinc oxide”
by Dobrina Ivanova, Hristo Kolev, Bozhidar I. Stefanov and Nina Kaneva.
Manuscript ID: catalysts-3163142
Dear Editors,
Thank you very much for the fast review process as well as the Reviewers comments and recommendations. Taking in mind all advices and suggestions we made the necessary corrections in order to improve the manuscript.
The new changes in the paper text are pointed out in green color. Please, see our comments below.
Reviewer 1:
Degradation of tetracycline antibiotic by utilizing pure and rare earth modified ZnO powders via tribocatalysis
Overview
Increased usage of antibiotics has become a major concern for environmentalists. This not only increases antibiotic resistance among microorganisms but also causes irreparable damage to living organisms and environment alike. The present work focuses on the degradation of doxycycline antibiotic by using rare earth (La/Eu) modified ZnO powders. Additionally, the effect of rpm was studied.
Major Comments
- Abstract should be re-written emphasizing the results. Introduction to the topic is too long.
The abstract has been re-written accordingly.
- Line 53 to 65: Introduction should focus on the application in removal or degradation of antibiotics.
We are thankful to this remark, however, there are only a few studies, which demonstrate (among other model pollutants), the degradation of antibiotics, and both focus on tetracycline. We have added the two references, after the mentioned paragraph.
- Line 88: From SEM images it is difficult know the size of the sample. How did authors determine the average particle size? If the particles are in “μm”, why “nanocomposites” word is used to describe them? Include SD as well.
In the revised version a particle-size analysis was carried on the particles shown in the SEM images and size-distributions are added. The materials are polycrystalline, as seen via TEM and XRD analysis, hence the term “nanocomposite” was used. The particles seen on the SEM images are larger agglomerates of multiple crystallites that cannot be discerned at the magnifications used.
- Figure 2, 3, 5: Include pure ZnO as well.
A TEM micrograph for the pure ZnO was added in Figure 3. Figure 5 includes the pure ZnO overlayed. We have not done EDX analysis on the pure ZnO, because it was commercially sourced, hence EDX was only used to confirm the Eu/La content before submitting the samples for the more-expensive and time-consuming XPS measurements.
- Table 1: Both Eu and La were too low. What is the basis for choosing 2 mol% concentration during preparation? Why not increase the concentration of Eu/La?
The basis for choosing the 2 mol.% concentration was founded on our previous experience with RE-modified ZnO photocatalysts, where this concentration was enough for a measurable effect, however, higher concentrations could lead to a decrease in activity. Additionally, while ZnO is quite inexpensive material, the RE-oxides are expensive.
- Figure 5: (a) Values 400, 440, 622, 200 were even observed in ZnO and ZnO/Eu. Similar can be observed in (b) as well. (b) no peak can be observed at 100, 102, 103. So, it appears XRD is inconclusive.
We apologize for this discrepancy. The peaks positions for the RE-oxides were marked only to indicate where they are expected in Figure 5. Given the low concentration of the secondary phase and the strong reflections from the crystalline ZnO it is not observed readily in the XRD due to the low sensitivity of the method (> 2 mol.%). This is why the XPS analysis was carried to confirm the presence of the respective RE-oxides.
- Figure 7: Should include SD as well.
The kinetic plots for the C/C0 and pseudo first order kinetics were updated accordingly.
- Line 252: States significant difference in bandgap values, whereas Figure 9 (Tauc plots) shows similar values.
The Duffy model assumes a stoichiometric mixed oxide (i.e. a ZnO/REO, where Zn:RE are equimolar). Due to the large differences in bandgaps between the RE oxide and the ZnO a significant difference, is thus, predicted. In the case of our study only 2 mol.% of the secondary RE phase is introduced. Nevertheless, the discussion in this rather simplistic approach (by Duffy) predicts qualitatively the small bandgap shifts in the resulting doped ZnO.
- Section 2.3.: Entire section i.e., plausible mechanism is theoretical. Authors should perform the “scavengers determination assay” to discuss about the role of different radicals or their involvement.
The mechanism of the tribocatalytic action is still under development. We based this section on literature, since elucidating the mechanism is quite out the scope of our study (and it is a new field for our group, which is much more versed in photocatalysis). Since tribocatalysis is a rather novel technique we believe that some discussion on the current understanding of its action is beneficial for an unfamiliar reader. Apart from this – we are grateful for the suggestion and can gladly confirm that a scavenger assay was carried with ascorbic acid and isopropanol, to act as a superoxide / hydroxide radical scavengers and the results are included in the revised version.
- Section 2.4.: Why not increase the rpm further?
The rpm effect investigation was carried to confirm (even to ourselves) that it is the mechanical action that is responsible for the DC removal and not simple adsorption. 500 rpm was decided as a maximum stirring rate, to avoid spattering of the solution, and also to prevent radical increases in the reaction temperature within the 24 hours duration of the experiment.
- Figure 10: Should include SD.
SD was added to the kinetic plots, similarly to Figure 7, in the revised version.
- Line 217: what is the basis for choosing pseudo-first order kinetics?
Pseudo-first order kinetics is qualitatively derived from the Langmuir-Hinshelwood adsorption model and is a kinetic model often employed in photo- and nowadays in tribocatalysis, due to its suitability for heterogeneous catalysts, where the limiting step is usually the adsorption of the model pollutant . References have been added to support the choice.
- Line 374: Where is UV-Vis spectra?
The UV vis spectra of DC removal has been added in the revised version as Figure 7.
Minor Comments
- Line 55: No need to state year. Ref inline journal format.
We have deleted the year in line 55 as indicated by the reviewer.
- Line 335 to 343: Should be moved to Introduction section.
we moved the side effects (Line 335 to 343) of the antibiotic in the introduction.
- Line 337, 339: Abbreviation and full name of doxycycline (DC) are randomly used. Should state “DC” after the first mention and abbreviation of doxycycline.
After the first mention and the abbreviation of doxycycline in the summary, everywhere in the article we corrected and the antibiotic is with its abbreviation only.
- Line 349: 100 ℃. Space should be between value and unit. Check the same in entire manuscript.
We put the space throughout the manuscript, as well as a space between the number and the %.
- Line 345, 352: RE or RE3+. Should use uniform representation. Check the same in entire manuscript.
We have corrected and used uniform representation throughout the article.
- Section 3: Use sub-headings accordingly.
The section was split into subheadings accordingly.
- Line 379, 382, 386, 392: “tribocatalytic”, “magnetically”, “intervals”, “decomposition”.
We have corrected the words indicated by the reviewer.
- Line 385: 300 rpm. No need to state rpm in full.
The necesseray correction has been made.
- Line 396: Reference for formula used.
A reference has been added.
- Line 405: “Eu3+ and La3+”.
The oxidation states were added.
Remark
The manuscript should be categorized properly. Introduction should be rearranged and should state the objectives more clearly. Should check for language mistakes.

Reviewer 2 Report
Comments and Suggestions for Authors
1. I suggest the authors have to revise the title, which can not indicate the current designing idea. Such as the modulation of degradation efficiency by dropping rare earth…?
2. Please provide the mapping data.
3. Why not discuss degradation efficiency on the Eu/La-ZnO?
4. The trapping experiment could be done, pls cite and refer to the refs, such as Molecules 2023, 28, 6848; J. Mol. Struct., 2024, 1312,138501. Also done the EPR for confirming the radical group for .OH or .O2-
5. Please provide the recycling number for the degradation.
Author Response
The manuscript: “Enhanced tribodegradation of a tetracycline antibiotic by rare-earth modified zinc oxide”
by Dobrina Ivanova, Hristo Kolev, Bozhidar I. Stefanov and Nina Kaneva.
Manuscript ID: catalysts-3163142
Dear Editors,
Thank you very much for the fast review process as well as the Reviewers comments and recommendations. Taking in mind all advices and suggestions we made the necessary corrections in order to improve the manuscript.
The new changes in the paper text are pointed out in green color. Please, see our comments below.
Reviewer 2:
- I suggest the authors have to revise the title, which can not indicate the current designing idea. Such as the modulation of degradation efficiency by dropping rare earth…?
We changed the title: “Enhanced tribodegradation of a tetracycline antibiotic by rare-earth modified zinc oxide”.
- Please provide the mapping data.
We added the mapping data of rare earth modified samples, new Figure 2c.
- Why not discuss degradation efficiency on the Eu/La-ZnO?
We thank the reviewer for the recommendation. This is a good and interesting idea for upgrading a three-component system. But this system it offers is not the focus of our presented research. We promise to deepen this research and verify the effectiveness of the particular sample- Eu/La-ZnO.
- The trapping experiment could be done, pls cite and refer to the refs, such as Molecules 2023, 28, 6848; J. Mol. Struct., 2024, 1312,138501. Also done the EPR for confirming the radical group for .OH or .O2-
We additionally performed 6 new tribocatalytic experiments to provide the trapping test, new Figure 10. We have cited the recommended articles:
[41] Zhao, J.; Dang, Z.; Muddassir, M.; Raza, S.; Zhong, A.; Wang, X.; Jin, J. A New Cd(II)-Based Coordination Polymer for Efficient Photocatalytic Removal of Organic Dyes. Molecules 2023, 28, 6848.
[42] Xiang, R.; Zhou, C.; Liu, Y.; Qin, T.; Li, D.; Dong, X.; Muddassir, M.; Zhong, A. A new type Co(II)-based photocatalyst for the nitrofurantoin antibiotic degradation. J. Mol. Struct. 2024, 1312,138501.
- Please provide the recycling number for the degradation.
We additionally performed 9 new tribocatalytic experiments to provide the recycle time at the three different stirring rates. Please refer the reviewer to the data presented in the new Figure - Figure 12.

Round 2
Reviewer 2 Report
Comments and Suggestions for Authors
it could be accept based on its current form.